# Ability of Carotid Corrected Flow Time to Predict Fluid Responsiveness in Patients Mechanically Ventilated Using Low Tidal Volume after Surgery

**DOI:** 10.3390/jcm10122676

**Published:** 2021-06-17

**Authors:** Seungho Jung, Jeongmin Kim, Sungwon Na, Won Seok Nam, Do-Hyeong Kim

**Affiliations:** Department of Anesthesiology and Pain Medicine, Anesthesia and Pain Research Institute, Yonsei University College of Medicine, Seoul 03722, Korea; jungshme@yuhs.ac (S.J.); ANESJEONGMIN@yuhs.ac (J.K.); NSWKSJ@yuhs.ac (S.N.); GALAXY995@yuhs.ac (W.S.N.)

**Keywords:** carotid artery, Doppler ultrasound, flow time, fluid therapy, tidal volume

## Abstract

Predicting fluid responsiveness in patients under mechanical ventilation with low tidal volume (VT) is challenging. This study evaluated the ability of carotid corrected flow time (FTc) assessed by ultrasound for predicting the fluid responsiveness during low VT ventilation. Patients under postoperative mechanical ventilation and clinically diagnosed with hypovolemia were enrolled. Carotid FTc and pulse pressure variation (PPV) were measured at VT of 6 and 10 mL/kg predicted body weight (PBW). FTc was calculated using both Bazett’s (FTcB) and Wodey’s (FTcW) formulas. Fluid responsiveness was defined as a ≥15% increase in the stroke volume index assessed by FloTrac/Vigileo monitor after administration of 8 mL/kg of balanced crystalloid. Among 36 patients, 16 (44.4%) were fluid responders. The areas under the receiver operating characteristic curves (AUROCs) for the FTcB at VT of 6 and 10 mL/kg PBW were 0.897 (95% confidence interval [95% CI]: 0.750–0.973) and 0.895 (95% CI: 0.748–0.972), respectively. The AUROCs for the FTcW at VT of 6 and 10 mL/kg PBW were 0.875 (95% CI: 0.722–0.961) and 0.891 (95% CI: 0.744–0.970), respectively. However, PPV at VT of 6 mL/kg PBW (AUROC: 0.714, 95% CI: 0.539–0.852) showed significantly lower accuracy than that of PPV at VT of 10 mL/kg PBW (AUROC: 0.867, 95% CI: 0.712–0.957; *p* = 0.034). Carotid FTc can predict fluid responsiveness better than PPV during low VT ventilation. However, further studies using automated continuous monitoring system are needed before its clinical use.

## 1. Introduction

The precise assessment of preload responsiveness to guide fluid administration is extremely important for improving the outcomes of critically ill patients. Insufficient intravascular volume can result in decreased cardiac output, leading to tissue hypoxia [1]. On the other hand, fluid overload can have deleterious effects, such as pulmonary edema, cardiac failure, impaired bowel function and even mortality [2,3]. However, determining the optimal volume status and fluid responsiveness in patients in critical care settings is challenging.

Pulse pressure variation (PPV) is one of the more widely used predictors of fluid responsiveness in patients under positive pressure ventilation [4,5]. PPV is based on the heart-lung interaction and needs enough intrathoracic pressure changes to predict preload responsiveness. Therefore, when using PPV, a tidal volume (VT) of >8 mL/kg predicted weight (PBW) is an essential condition for an accurate prediction [6,7]. Meanwhile, using a target VT of <6 mL/kg PBW has been proven to reduce mortality in patients with acute respiratory distress syndrome (ARDS), and is now generally used in critical care settings [8,9,10].

Recently, corrected flow time (FTc) measured in the carotid artery has been reported as a reliable predictor of fluid responsiveness during spontaneous breathing [11,12]. Additionally, one study reported that the variation of intrathoracic pressure during respiration did not significantly affect the measured carotid FTc [13]. Recently, Barjaktarevic and colleagues showed that the change in carotid FTc after passive leg raising (PLR) maneuver was able to predict fluid responsiveness status in shock patients, including those receiving mechanical ventilation [14]. Therefore, carotid FTc could be a promising predictor of fluid responsiveness in patients receiving mechanical ventilation with low VT.

This study aimed to evaluate whether carotid FTc measured by Doppler ultrasound could be a reliable predictor of fluid responsiveness in mechanically ventilated patients with low VT in critical care settings. We also compared the predictability of carotid FTc and PPV measured during low VT ventilation with each of the parameters measured when increasing the VT to 10 mL/kg PBW.

## 2. Materials and Methods

### 2.1. Study Population

The study protocol was approved by the Institutional Review Board of the Yonsei University Health System, Seoul, South Korea (no. 4-2019-0848, dated 21 October 2019) and registered in ClinicalTrials.gov (NCT04139031, Principal Investigator: Do-Hyeong Kim). After receiving written informed consent, we enrolled 37 patients from October 2019 to September 2020.

This prospective single-center study was performed in the surgical intensive care unit (ICU) of a university hospital. The inclusion criteria were as follows: patients aged >19 years who were planned to be transferred to the ICU for mechanical ventilation after surgery, clinically diagnosed with hypovolemia, and planned to be resuscitated with fluid administration by the attending physicians. Hypovolemic status was diagnosed based on at least one of the clinical signs presented in Appendix A [1]. The exclusion criteria were as follows: body mass index of >40 or <15 kg/m^2^; common carotid artery stenosis, of >50% which was previously diagnosed (by conventional angiography, computed tomographic angiography, magnetic resonance angiography or duplex ultrasonography) or newly detected during the study period; cardiac rhythm other than sinus rhythm observed during the study period; moderate to severe valvular heart disease detected by preoperative echocardiography; left ventricular ejection fraction of <50%; right ventricular failure; moderate to severe chronic obstructive pulmonary disease; pulmonary mean arterial pressure of >25 mmHg; suspected or diagnosed increased intracranial pressure; pregnancy; patients who were unable to read the consent form (e.g., illiterate, foreigner, etc.); and need for vasopressor infusion to maintain normal blood pressure before being transferred to the ICU. Furthermore, we excluded patients whom the ultrasound probe could not reach the carotid artery because of an overlapping position of the surgical wound and exam site.

### 2.2. Postoperative Management

After surgery, 0.05 mg/kg of midazolam and 0.6 mg/kg of rocuronium were administered to maintain the sedation status until transferred to the ICU. Upon arrival to the ICU, the IntelliVue MX700 monitor (Philips Medical Systems, Böblingen, Germany) was used to monitor the hemodynamic variables, including PPV, using the previously described algorithms [15]. A radial arterial catheter was connected to the FloTrac sensor in conjunction with the Vigileo platform (software version 1.9, Edwards Lifesciences, Irvine, CA, USA) for stroke volume index (SVI) monitoring. The initial VT for mechanical ventilation was set at 6 mL/kg PBW. The initial ventilator settings were as follows: fraction of inspired oxygen, 0.4; ratio of inspiration and expiration, 1:2; positive end-expiratory pressure (PEEP), 5 cmH_2_O; and respiratory rate, 15−18 times per minute. All patients were placed in the 30° semirecumbent position.

### 2.3. Study Protocol

The study design is illustrated in Figure 1. When the patient reached a hemodynamic steady state, the baseline hemodynamic and respiratory variables, including heart rate, mean arterial pressure, SVI, carotid FTc (FTc_6_), PPV at a VT of 6 mL/kg PBW (PPV_6_), driving pressure (plateau pressure − PEEP) and compliance of the respiratory system were recorded. After recording this data set, the VT setting was increased from 6 to 10 mL/kg PBW. The same data set, including carotid FTc (FTc_10_) and PPV at a VT of 10 mL/kg PBW (PPV_10_), was measured five minutes after increasing the VT. Then, we reset the VT to 10 mL/kg PBW and administered 8 mL/kg ideal body weight of balanced crystalloid solution (Plasma Solution A, CJ Pharmaceutical, Seoul, Korea) over 10 min. Five minutes after the completion of fluid administration, the above-mentioned parameters were recorded again. No vasoactive medication was administered during the assessment period, and all measurements were obtained when the blood pressure and heart rate did not fluctuate. Fluid responsiveness was defined as a ≥15% increase in SVI after fluid administration.

### 2.4. Carotid Ultrasonography

The FTc was assessed using an ultrasound device (SonoSite M-Turbo; SonoSite Inc., Bothell, WA, USA) as previously described by Blehar and colleagues [16]. A 6.0−13.0 MHz linear array transducer (HFL38xp; SonoSite Inc., Bothell, WA, USA) was used to perform a pulsed wave-Doppler tracing of flow through the common carotid artery, and blood flow waveforms were captured. The cycle time was obtained by measuring the interval between heartbeats at the beginning of the systolic upstroke. The flow time was obtained by measuring the interval between the systolic upstroke and dicrotic notch in the 10th of microsecond increments. The corrected blood flow time was calculated by using the Bazett’s (FTcB) and Wodey’s (FTcW) formulas. Bazett’s formula is calculated by dividing the flow time by the square root of the cycle time. Wodey’s formula is calculated according to the following: flow time measured + [1.29 × (heart rate − 60)] [17]. All three FTc measurements were performed in real time by a single pre-trained examiner, while another independent investigator noted the other haemodynamic and respiratory parameters. After the study protocol, a second independent examiner who was blinded to first measurement re-assessed FTc using stored unprocessed ultrasound images to assess inter-observer variability.

### 2.5. Statistical Analysis

A previous study reported that the area under the receiver operating characteristic curve (AUROC) of FTc measured in the descending aorta to predict fluid responsiveness was 0.82 [18]. We assumed that the AUROC of carotid FTc was 0.80, a rather lower value. The sample size calculation showed that at least 33 patients were needed to detect a difference of 0.30 between the AUROCs of the carotid FTc (0.80) and the null hypothesis (0.50), with a power of 0.9 and a two-tailed type I error of 0.05, assuming a fluid responsiveness incidence of 56% in mechanically ventilated patients [19]. To allow for a possible 10% dropout rate, 37 patients were finally required.

The continuous variables were analysed by using the independent t-test or Mann–Whitney U test, according to the results of the Shapiro-Wilk test for normality. The categorical variables were analysed by the chi-square test or Fisher’s exact test. The binary data are presented as numbers (%), while the continuous data are presented as the means and standard deviations if normally distributed, or as medians and interquartile ranges if otherwise. The AUROC was calculated to measure the ability of the indices to predict fluid responsiveness. A comparison of the AUROCs was performed using the non-parametric technique proposed by DeLong and colleagues [20]. The optimal cut-off value was determined by maximizing the Youden index. Using bootstrap methodology with 1000 multiple samples, 95% confidence intervals (CIs) of the best threshold were determined as the grey zone [21]. A Spearman’s rank correlation coefficient was used to test the relationship between the percent changes in SVI and FTc from baseline to after fluid loading. The inter-observer reproducibility was assessed in all data sets by calculating an intraclass correlation coefficient (ICC) and a coefficient of variation (CV). The inter-observer agreement in estimating FTc was tested using the Bland-Altman plot. All analyses were performed using the Statistical Package for the Social Sciences (SPSS) version 25 software (IBM Corp., Armonk, NY, USA), R version 3.5.3 (The R Foundation for Statistical Computing, Vienna, Austria) and MedCalc version 19.5.1 (MedCalc, Ostend, Belgium). Statistical significance was set as *p* < 0.05.

## 3. Results

### 3.1. Patient Characteristics

Of the 52 patients assessed for eligibility, 37 were enrolled. One patient was dropped out because of unreliable hemodynamic values owing to the malfunctioning of the arterial line monitoring during the study. Thus, 36 patients were included in the final analysis (Appendix A). No significant differences were found in patient characteristics and anesthetic details between responders (n = 16) and non-responders (n = 20), except for the total anesthesia time, which was longer among responders than among non-responders (Table 1).

### 3.2. Hemodynamic Changes and Respiratory Variables

Changes in hemodynamic and respiratory variables at each point of the study are shown in Table 2. With increased VT, the driving pressure significantly increased, while the respiratory system compliance was not significantly changed. SVI was significantly lower in responders than in non-responders at baseline, although SVI was not significantly different between the two groups after fluid loading. In both responders and non-responders, fluid loading significantly increased SVI. FTcB and FTcW were significantly lower in responders than in non-responders at the baseline. FTcB and FTcW were significantly lower in responders than in non-responders at baseline, and when the VT was increased to 10 mL/kg PBW, PPV was significantly higher in responders. Fluid administration significantly increased FTcB and FTcW only in responders. The percent changes in FTcB and FTcW after the fluid challenge correlated with the percent change in SVI (rho = 0.654, 95% CI: 0.414–0.809; *p* < 0.0001, rho = 0.656, 95% CI: 0.417–0.810; *p* < 0.0001, respectively) (Appendix A).

### 3.3. Prediction of Fluid Responsiveness

The AUROCs for the FTc_6_B and FTc_10_B were 0.897 (95% CI: 0.750–0.973; *p* < 0.0001) and 0.895 (95% CI: 0.748–0.972; *p* < 0.0001), respectively (Table 3 and Figure 2). The optimal cut-off values of the FTc_6_B and FTc_10_B for predicting fluid responsiveness were 338.5 ms and 345.1 ms, respectively. The AUROCs for the FTc_6_W and FTc_10_W were 0.875 (95% CI: 0.722–0.961; *p* < 0.0001) and 0.891 (95% CI: 0.744–0.970; *p* < 0.0001), respectively. The optimal cut-off values of the FTc_6_W and FTc_10_W were 325.8 ms and 335.8 ms, respectively. There were no significant differences in the AUROCs both between the FTc_6_B and FTc_10_B and between the FTc_6_W and FTc_10_W. The AUROCs for the PPV_6_ and PPV_10_ were 0.714 (95% CI: 0.539–0.852; *p* = 0.0139) and 0.867 (95% CI: 0.712–0.957; *p* < 0.0001), respectively. However, the predictive accuracy for fluid responsiveness of PPV_6_ was significantly lower than that of PPV_10_ (*p* = 0.034).

The inter-observer reproducibility for estimating FTcB and FTcW was excellent with an ICC of 0.98 (95% CI: 0.96–0.98) and a CV of 1.2% and an ICC of 0.98 (95% CI: 0.97–0.99) and a CV of 1.1%, respectively. Using Bland-Altman analysis for testing inter-observer agreement in estimating FTcB and FTcW, the mean biases were −0.42 ms (with 95% limits of agreement [LOA] between −11.80 and 12.65 ms) and −0.44 ms (with 95% LOA between −10.11 and 10.99 ms), respectively (Appendix A).

## 4. Discussion

This study demonstrated that carotid FTc measured by Doppler ultrasound is a valid and reliable predictor for determining fluid responsiveness in patients receiving positive pressure mechanical ventilation in either a low VT or an increased VT setting. In contrast, PPV showed a limited accuracy in the presence of low VT than that of in the presence of increased VT.

After reported by the ARDS network, a low VT ventilation strategy is known to be the basic principle to reduce ventilator-induced lung injury [8,10]. Therefore, mechanical ventilation using low VT is widely recommended in the perioperative period and critical care settings [9,22]. Low VT is usually considered as a VT of <6 mL/kg PBW. PPV in those circumstances is reported as less reliable than when the VT is at least 8 mL/kg PBW. Consistent with a previous study, PPV measured in a low VT setting showed a lower accuracy in predicting fluid responsiveness (AUROC = 0.714) than that measured in an increased VT setting (AUROC = 0.867) in this study. To overcome the low performance of PPV in the presence of a low VT, some challenges have been reported. In a study, it was attempted to adjust the PPV value according to the respiratory changes in pleural pressure measured by an esophageal catheter [23]. Another way is to increase the VT transiently to observe the changes in PPV [24]. However, these strategies need additional invasive monitoring or ventilator setting changes to improve the performance of PPV.

The rationale of FTc is based on the usefulness of the Doppler waveform in the descending thoracic aorta about guiding fluid management [25,26]. FTc measured in other superficial arteries, such as the carotid artery, made it practical to use this technique for guiding fluid resuscitation [11,12,16]. Theoretically, static parameters such as carotid FTc are known to be less affected by respiration than dynamic parameters. The doctor and colleagues reported that the variations in the intrathoracic pressure during respiration did not significantly affect carotid FTc [13]. However, few studies have reported the accuracy of carotid FTc under low VT setting in mechanical ventilation. In our study, carotid FTc under a low VT setting was found to be as good as PPV with a VT of 10 mL/kg PBW for predicting fluid responsiveness. As mentioned above, using a low VT for mechanical ventilation can limit the predictability of PPV. Under these circumstances, using carotid FTc for predicting fluid responsiveness could be a good alternative method that does not require manipulation of the VT setting or other additional invasive procedures.

Our study estimated the predictability of fluid responsiveness by using the absolute value of FTc. However, there are some limitations of using the FTc value, as it can be affected not only by preload but also by cardiac contractility or afterload variability. We excluded patients who had a history of heart failure. Additionally, we excluded patients receiving vasopressor infusion, and the PEEP level was fixed during measurement. These conditions might minimise the effects other than preload to FTc; thus, FTc showed high predictability in our study. There are some reports on the predictability of changes in carotid FTc for fluid responsiveness [14,27]. Due to the limitations of the absolute FTc value mentioned previously, Barjaktarevic and colleagues used the changes in carotid FTc by PLR maneuver to predict fluid responsiveness in patients with shock and using vasopressor [14]. As afterload can be affected by vasopressor infusion, the changes in carotid FTc may be more useful as a predictor of fluid responsiveness than the absolute FTc value under these conditions.

Our study has some limitations. First, we assumed that the variations in intrathoracic pressure during respiration did not significantly affect the carotid FTc. In addition, there were little differences between FTc_6_ and FTc_10_. However, a recent study showed that the respiratory cycle could induce a variation in FTc [28]. This could affect the results of our study. A study for the evaluation of the FTc variation during the respiratory cycle is needed. Further studies in patients with poor lung compliance or large intrathoracic pressure change may result in different results. Second, the thermodilution technique via pulmonary artery catheter (PAC) insertion was too invasive for our patients. Therefore, we used the FloTrac sensor to measure the change in SVI. The Flotrac/VigileoTM system is a minimally invasive cardiac output monitoring system, which has a blood flow sensor connected to the patient’s arterial line. It calculates cardiac output by using individual demographic data and arterial pressure waveform analysis. Cardiac output measured by Flotrac showed acceptable agreement with thermodilution technique [29]. It is an uncalibrated device, and showed inaccuracies in patients with severe arrhythmia, severe aortic valve regurgitation, and in those using high dose vasopressor [30]. However, the Flotrac system showed tight cross-correlations with other cardiac output measuring devices in monitoring the change of cardiac output, and is reported as a reliable device [31]. Therefore, it has been used to monitor the cardiac output in several studies [27,32,33]. Third, this study was conducted in a small population. Our study was performed by a single ultrasonography examiner in a surgical ICU, which limited the enrollment of a large number of patients. Our findings need to be reproduced in a study with a larger number of subjects. Finally, the measurement of FTc in our study was performed manually by two pre-trained experts using captured images. Although excellent inter-observer reproducibility for the measurement of FTc was noted in our study, ultrasound images were captured only once by a single operator, and we could not rule out the possibility of inter-observer variability while acquiring the images. We also could not report intra-observer variability because the ultrasound image was captured only once at each point. Recently, a carotid Doppler patch has been developed with which a continuous waveform can be monitored [34]. A computer-programmed algorithm for detecting the onset, systolic peak and dicrotic notch in the arterial waveform has been also reported [35]. We may continuously monitor FTc, and also minimise human measurement error after adequate validations of these technologies.

## 5. Conclusions

In conclusion, the carotid artery FTc measured by Doppler ultrasound was found to be a useful predictor for evaluating fluid responsiveness in mechanically ventilated patients using low VT. Unlike PPV, which showed low predictability during low VT, carotid FTc showed high predictability during both low and high VT mechanical ventilation. The wide grey zone, which can limit the clinical use of FTc, should be overcome in larger clinical trials.

## Figures and Tables

**Figure 1 jcm-10-02676-f001:**
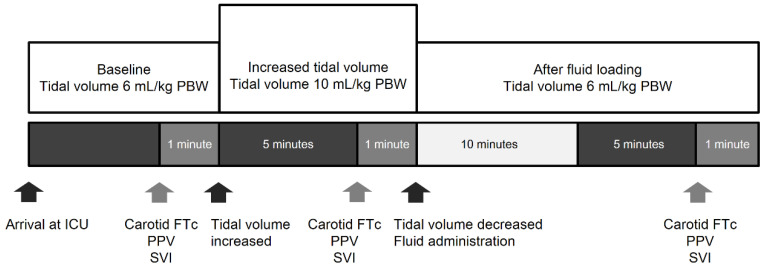
Study design. The arrows indicate the time points at which interventions or measurements were made. PBW = predicted body weight; PPV = pulse pressure variation; SVI = stroke volume index.

**Figure 2 jcm-10-02676-f002:**
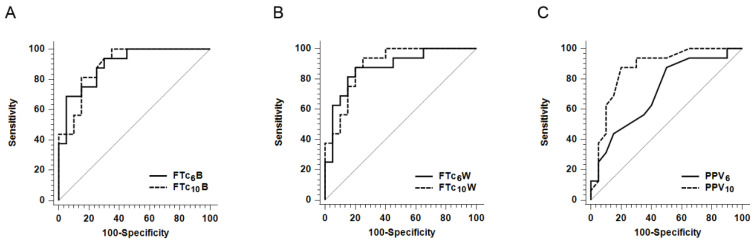
Receiver operating characteristic curves comparing the ability of various parameters to predict the fluid responsiveness. (**A**) FTc6B vs. FTc10B. (**B**) FTc6W vs. FTc10W. (C) PPV6 vs. PPV10. The area under the receiver operating characteristic curve for PPV6 was only significantly lower than that of PPV10 (*p* = 0.034). FTc6B = corrected flow time in the carotid artery calculated by Bazett’s formula at tidal volume (VT) of 6 mL/kg predicted body weight; FTc10B = corrected flow time in the carotid artery calculated by Bazett’s formula at VT of 10 mL/kg predicted body weight; FTc6W = corrected flow time in the carotid artery calculated by Wodey’s formula at VT of 6 mL/kg predicted body weight; FTc10W = corrected flow time in the carotid artery calculated by Wodey’s formula at VT of 10 mL/kg predicted body weight; PPV6 = pulse pressure variation at VT of 6 mL/kg predicted body weight; and PPV10 = pulse pressure variation at VT of 10 mL/kg predicted body weight.

**Table 1 jcm-10-02676-t001:** Patient characteristics and details of anesthesia.

	Overall (*n* = 36)	Responders (*n* = 16)	Non-Responders (*n* = 20)	*p*-Value
Female, *n* (%)	14 (38.9)	5 (31.2)	9 (45.0)	0.619
Age (year)	55.0 ± 11.5	52.8 ± 13.0	56.9 ± 10.2	0.295
Height (cm)	165.6 ± 8.2	167.6 ± 6.6	164.0 ± 9.2	0.200
Weight (kg)	62.9 ± 12.0	65.7 ± 12.0	60.6 ± 11.9	0.210
BMI (kg/m^2^)	22.9 ± 2.9	23.3 ± 3.2	22.4 ± 3.0	0.411
ASA physical status (I/II/III)	13/18/5	5/10/1	8/8/4	0.343
APACHE II score at ICU admission	18.1 ± 4.9	17.4 ± 5.9	18.7 ± 4.0	0.469
Diagnosis, *n* (%)				0.550
Head and neck cancer	31 (86.1)	13 (81.3)	18 (90.0)	
Facial bone fracture	1 (2.8)	0 (0)	1 (5.0)	
Intra-abdominal cancer	3 (8.3)	2 (12.5)	1 (5.0)	
Cervical disc herniation	1 (2.8)	1 (6.3)	0 (0)	
Comorbidities, *n* (%)				
Hypertension	12 (33.3)	7 (43.8)	5 (25.0)	0.406
Diabetes mellitus	5 (13.9)	2 (12.5)	3 (15.0)	>0.999
Coronary artery disease	1 (2.8)	0 (0)	1 (5.0)	>0.999
Medications, *n* (%)				
Calcium channel blockers	9 (25.0)	5 (31.3)	4 (20.0)	0.699
β-blockers	2 (5.6)	1 (6.3)	1 (5.0)	>0.999
Angiotensin receptor blockers	8 (22.2)	5 (31.3)	3 (6.0)	0.446
Statin	7 (19.4)	2 (12.5)	5 (25.0)	0.605
Fluid volume administered (mL/h) *	282.7 [243.2−344.8]	282.1 [222.4−376.3]	282.7 [245.1−335.7]	0.863
Urine output (mL/h) †	74.0 [60.9–136.2]	74.0 [63.0–122.9]	75.0 [59.6–154.8]	0.814
Blood loss (mL)	110.0 [50.0−350.0]	350.0 (75.0−650.0)	100.0 (50.0−160.0)	0.110
Amount of pRBCs transfused (mL)	0.0 [0.0−0.0]	0.0 [0.0−0.0]	0.0 [0.0−0.0]	0.199
Patients received vasopressors during surgery, *n* (%)	17 (47.2)	7 (43.8)	10 (50.0)	0.970
Anesthesia time (min)	455.0 [342.5−605.0]	510.0 [415.0−905.0]	390.0 [317.5−492.5]	0.012

The data are presented as the mean ± SD, median [IQR], or the number of patients (%). * Fluid volume administered, or the fluid volume administered per hour during anesthesia. † The urine output per hour during anesthesia. ASA = American Society of Anesthesiologists; APACHE II = Acute Physiology and Chronic Health Evaluation II; BMI = body mass index; ICU = Intensive Care Unit; IQR = interquartile range; pRBCs = packed red blood cells; SD = standard deviation.

**Table 2 jcm-10-02676-t002:** Hemodynamic and respiratory variables at baseline, during increased tidal ventilation, and after fluid loading.

	Baseline	Increased V_T_ Ventilation	After Fluid Loading
	V_T_, 6 mL/kg PBW	V_T_, 10 mL/kg PBW	V_T_, 6 mL/kg PBW
Heart rate (beats/min)			
Responders	91.1 ± 16.0	91.4 ± 16.3	82.9 ± 13.8 ‡,§
Non-responders	81.4 ± 13.7	82.5 ± 15.6	78.7 ± 12.4 §
Mean arterial pressure (mmHg)			
Responders	77.3 ± 18.4	72.3 ± 18.2 *,†	80.8 ± 15.8 §
Non-responders	87.3 ± 12.6	83.8 ± 11.6†	82.6 ± 10.3 ‡
Driving pressure (cmH_2_O)			
Responders	7.0 [7.0−9.0]	12.0 [10.5−13.0] †	7.0 [7.0−8.5] §
Non-responders	8.0 [6.0−8.0]	13.0 [10.5−14.0] †	8.0 [7.0−9.0] §
Respiratory system compliance (mL/cmH_2_O)			
Responders	57.0 [38.4−73.6]	59.0 [51.6−79.0]	59.2 [38.9−70.8]
Non-responders	50.9 [44.7−71.4]	54.5 [47.7−68.3]	52.3 [43.3−72.3]
Stroke volume index (mL/m^2^)			
Responders	36.5 [33.0−38.5] *	36.0 [31.0−39.0]	47.5 [41.0−52.0] ‡,§
Non-responders	41.5 [38.0−51.5]	40.0 [35.0−50.0] †	45.0 [38.0−53.0] ‡,§
FTcB (ms)			
Responders	329.3 ± 19.0*	328.9 ± 18.8 *	362.3 ± 20.9 ‡,§
Non-responders	363.5 ± 21.1	362.8 ± 20.0	366.7 ± 25.3
FTcW (ms)			
Responders	311.8 ± 18.3 *	310.1 ± 18.6 *	341.8 ± 25.9 ‡,§
Non-responders	344.3 ± 22.7	342.8 ± 21.5	348.3 ± 27.8
PPV (%)			
Responders	9.0 [7.0−13.0] *	16.0 [14.0−22.5] *,†	6.0 [5.0−9.5] *,‡,§
Non-responders	6.5 [5.0−9.0]	9.5 [7.0−13.0] †	5.0 [4.0−6.5] ‡,§

The data are presented as the mean ± SD or median [IQR]. * *p* < 0.05, fluid responders vs. fluid non-responders (comparison in columns). † *p* < 0.05, increased V_T_ ventilation vs. baseline (comparison in rows). ‡ *p* < 0.05, after fluid loading vs. baseline (comparison in rows). § *p* < 0.05, after fluid loading vs. increased V_T_ ventilation (comparison in rows). FTcB = corrected flow time in the carotid artery calculated by Bazett’s formula; FTcW = corrected flow time in the carotid artery calculated by Wodey’s formula; IQR = interquartile range; PBW = predicted body weight; PPV = pulse pressure variation; SD = standard deviation; and V_T_ = tidal volume.

**Table 3 jcm-10-02676-t003:** Diagnostic accuracy of various variables to predict fluid responsiveness.

	AUROC (95% CI)	*p*-Value	Cut-Off Value *	Grey Zone †	Patients in Grey Zone (%)	Sensitivity (%) (95% CI)	Specificity (%) (95% CI)
FTc_6_B	0.897 (0.750–0.973)	<0.0001	338.5 ms	317.5−350.7 ms	13 (36)	68.8 (41.3–89.0)	95.0 (75.1–99.9)
FTc_10_B	0.895 (0.748–0.972)	<0.0001	345.1 ms	316.6−355.6 ms	16 (44)	81.3 (54.4–96.0)	85.0 (62.1–96.8)
FTc_6_W	0.875 (0.722–0.961)	<0.0001	325.8 ms	322.9−349.4 ms	12 (33)	87.5 (61.7–98.4)	80.0 (56.3–94.3)
FTc_10_W	0.891 (0.744–0.970)	<0.0001	335.8 ms	328.7−339.4 ms	8 (22)	93.8 (69.8–99.8)	75.0 (50.9–91.3)
PPV_6_	0.714 (0.539–0.852)	0.0139	6%	5−9%	23 (64)	87.5 (61.7–98.4)	50.0 (27.2–72.8)
PPV_10_	0.867 (0.712–0.957)	<0.0001	13%	12−17%	14 (39)	87.5 (51.7–98.4)	80.0 (56.3–94.3)

* The optimal cut-off values were determined by maximizing the Youden index. † Grey zone: based on 1000 bootstrap samples, 95% confidence intervals of the best threshold were determined. AUROC = area under the receiver operating characteristic curve; CI = confidence interval; PBW = predicted body weight; V_T_ = tidal volume; FTc_6_B = corrected flow time in the carotid artery calculated by Bazett’s formula at a V_T_ of 6 mL/kg PBW; FTc_10_B = corrected flow time in the carotid artery calculated by Bazett’s formula at a V_T_ of 10 mL/kg PBW; FTc_6_W = corrected flow time in the carotid artery calculated by Wodey’s formula at a V_T_ of 6 mL/kg PBW; FTc_10_W = corrected flow time in the carotid artery calculated by Wodey’s formula at a V_T_ of 10 mL/kg PBW; PPV_6_ = pulse pressure variation at a V_T_ of 6 mL/kg PBW; and PPV_10_ = pulse pressure variation at a V_T_ of 10 mL/kg PBW.

## Data Availability

The datasets generated and/or analyzed during the current study are not publicly available due to our IRB policy but are available from the corresponding author upon reasonable request.

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
