# Peer review of "Ability of Carotid Corrected Flow Time to Predict Fluid Responsiveness in Patients Mechanically Ventilated Using Low Tidal Volume after Surgery"

_jcm, 2021, doi:10.3390/jcm10122676_

Round 1

Reviewer 1 Report

The authors should be commended for carrying out this interesting and important work.

They have studied fluid responsiveness in relatively, young, thin patients who were post-operative from head and neck cancer surgery. They have appropriately excluded patients in whom pulse pressure variation is an invalid measure of fluid responsiveness (e.g., arrhythmia, severe heart failure, respiratory effort, etc.).

They ask whether the corrected flow time measured by carotid Doppler can predict fluid responsiveness in these patients receiving both low and high tidal volume and compare this to pulse pressure variation. The reference standard is a 15% change in stroke volume following fluid challenge as measured by Flotrac.

There are a few issues that need to be further addressed prior to acceptance for publication in this journal.

  1. The authors note that they use FTc here as a ‘static’ measure because they rely on absolute FTc as their diagnostic measure rather than change in FTc as Barjakterevic et al. did. Yet it appears in the author’s data that changing FTc could be a valid surrogate for changing SV as responders had significantly lower FTc and significantly lower SV at baseline. Then when fluid was given, both SV and FTc changed (increased) significantly in responders. This large delta held true at both 6 and 10 tidal volume conditions. It appears from this raw data that FTc change following IV fluids actually tracks SV change quite well? For example, based on FTcW, responders increased FTc by about 10% after fluids, while non-responders had, on average, 1-2% change after fluids. 

-Can the authors comment on how changing FTc might track changing SV (see also, http://doi.org/10.1186/s40635-020-00339-7); was this analysis performed? How does this change in FTc relate to this relevant publication (http://dx.doi.org/10.1016/j.amjms.2017.09.006) which used an identical gold standard?

  1. the authors report the results of Doctor et al. as evidence that FTc is not affected by the respiratory cycle. If FTc is a surrogate for SV, then one would expect FTc to vary about the respiratory cycle as SV does. Note that older physiological data (10.1016/0002-8703(61)90403-3) and this more recent healthy volunteer data (https://doi.org/10.1186/s40635-020-00339-7) suggest that FTc varies with respiration, like SV.

-Could the authors comment on this and also make clear *where in the respiratory cycle were their Doppler measures taken?  How does this relate to how the SV value from Flotrac was obtained in the respiratory cycle?

  1. Line 323 should reflect a large change in trans-pulmonary* pressure in severe pulmonary ARDS, rather than intrathoracic* pressure. Intrathoracic pressure (i.e., pleural pressure) would change more with extrapulmonary ARDS and/or obesity

  1. the format of table 3 makes it difficult to follow. I suggest making it more pleasing by spacing it better, this table contains very important information that should be easy for the reader to extract.

  1. can the authors comment on limitations between Bazett and Wodey’s equations, for example how each of them handles extremes of heart rates? Is this a factor for the heart rates reported in the author’s work?

  1. can the authors comment on the heart rate-to-respiratory rate ratio and how this affects pulse pressure variation as a measure of fluid responsiveness? Was this a factor here?

  1. lines 241-243 and 346-350 appear to be vestiges from the journal’s manuscript template; please cut.

Author Response

Response to Reviewer 1 Comments
The authors should be commended for carrying out this interesting and important work.

They have studied fluid responsiveness in relatively, young, thin patients who were post-operative from head and neck cancer surgery. They have appropriately excluded patients in whom pulse pressure variation is an invalid measure of fluid responsiveness (e.g., arrhythmia, severe heart failure, respiratory effort, etc.).

They ask whether the corrected flow time measured by carotid Doppler can predict fluid responsiveness in these patients receiving both low and high tidal volume and compare this to pulse pressure variation. The reference standard is a 15% change in stroke volume following fluid challenge as measured by Flotrac.

There are a few issues that need to be further addressed prior to acceptance for publication in this journal.

We would like to thank you for your time and effort in reviewing our manuscript and providing valuable comments, which have greatly helped us in revising the manuscript. We have carefully read your comments and have revised the manuscript. Our detailed responses to each comment are written below.

Point 1: The authors note that they use FTc here as a ‘static’ measure because they rely on absolute FTc as their diagnostic measure rather than change in FTc as Barjakterevic et al. did. Yet it appears in the author’s data that changing FTc could be a valid surrogate for changing SV as responders had significantly lower FTc and significantly lower SV at baseline. Then when fluid was given, both SV and FTc changed (increased) significantly in responders. This large delta held true at both 6 and 10 tidal volume conditions. It appears from this raw data that FTc change following IV fluids actually tracks SV change quite well? For example, based on FTcW, responders increased FTc by about 10% after fluids, while non-responders had, on average, 1-2% change after fluids. 

-Can the authors comment on how changing FTc might track changing SV (see also, http://doi.org/10.1186/s40635-020-00339-7); was this analysis performed? How does this change in FTc relate to this relevant publication (http://dx.doi.org/10.1016/j.amjms.2017.09.006) which used an identical gold standard?

Response 1
As you pointed out, measurement of the change in FTc tracked the changes in stroke volume in previous studies [1,2]. In our study, the percent changes in FTcB and FTcW after fluid challenge significantly correlated with the percent change in SVI (rho=0.654, 95% CI: 0.414-0.809, P < 0.0001; rho=0.656, 95% CI: 0.417-0.810, P < 0.0001, respectively). This is consistent with findings of previous studies. The study you have mentioned [1] was performed in spontaneous breathing patients and used the squat method as a preload modifying maneuver. They also used a wearable Doppler patch to estimate the changes of FTc. The study, which used the Flotrac monitor similar to that in our study[2], measured stroke volume rather than stroke volume index. They also used the PLR maneuver rather than fluid challenge. These points could explain the differences between the previous studies and our study. We planned this study to estimate the ability of the absolute FTc value as a point-of-care technique for predicting fluid responsiveness. However, as per your advice, we have added the Spearman’s correlation analysis in the Methods section, the results of the analysis in the Results section, and Supplemental Figure 2. 

Original sentence 
The optimal cut-off value was determined by maximising the Youden index. Using bootstrap methodology with 1000 multiple samples, 95% confidence intervals (CIs) of the best threshold were determined as the grey zone [21].
Revised sentence (2.5. Statistical Analysis section, Lines 178-180)
The optimal cut-off value was determined by maximizing the Youden index. Using bootstrap methodology with 1000 multiple samples, 95% confidence intervals (CIs) of the best threshold were determined as the grey zone [21]. A Spearman's rank correlation coefficient was used to test the relation-ship between the percent changes in SVI and FTc from baseline to after fluid loading.

Original sentence 
Fluid administration significantly increased FTcB and FTcW only in responders.

Revised sentence (Results section, Lines 219-223)
Fluid administration significantly increased FTcB and FTcW only in responders. The percent changes in FTcB and FTcW after the fluid challenge correlated with the percent change in SVI (rho=0.654, 95% CI: 0.414-0.809; P<0.0001, rho=0.656, 95% CI: 0.417-0.810; P<0.0001, respectively) (Supplemental Fig 2).

Point 2: the authors report the results of Doctor et al. as evidence that FTc is not affected by the respiratory cycle. If FTc is a surrogate for SV, then one would expect FTc to vary about the respiratory cycle as SV does. Note that older physiological data (10.1016/0002-8703(61)90403-3) and this more recent healthy volunteer data (https://doi.org/10.1186/s40635-020-00339-7) suggest that FTc varies with respiration, like SV.

-Could the authors comment on this and also make clear *where in the respiratory cycle were their Doppler measures taken?  How does this relate to how the SV value from Flotrac was obtained in the respiratory cycle?

Response 2
As you mentioned, FTc and stroke volume could be affected by the respiratory cycle. We planned this study based on the assumption that the respiratory cycle has little effect on FTc measurements. This assumption was made based on the study by Doctor et al. [3] and a study that showed predictability of FTc in spontaneous breathing patients [4]. Additionally, to minimize the effect of the respiratory cycle on FTc, we tried to measure the FTc during the end-expiratory phase. However, we cannot ensure that the measurement of FTc was performed only during the end-expiratory phase because of the limited function of the ultrasound device which could not automatically synchronize the respiratory cycle with FTc measurement. The timing of measurement and respiratory cycle were synchronized manually. We recorded updated stroke volume index values right after the FTc measurement. Stroke volume measured by Flotrac was updated every 60 seconds, which also meant that the values could not be obtained exactly during the end expiratory phase only. These could be the limitations of our study. However, there are some differences between the study of Kenny et al and our study. That study was conducted in patients who were breathing spontaneously and were standing. Our study was in patients who were mechanically ventilated and placed in the 30° semirecumbent position. The tidal volume required for normal spontaneous breathing is usually higher than the tidal volume used in our study. These may explain the differences of mean baseline FTc values estimated between the two studies. In addition, the driving pressure measured in our study was 7.5 cmH2O (range, 6.0-8.5); therefore, the effect of the transmitted pressure to FTc and stroke volume could have been minimized [5, 6]. Furthermore, there were little differences between the FTc values measured at tidal volume of 6 and 10mL/kg predicted body weight. Evaluation of FTc variation during the respiratory cycle in patients using low tidal volume ventilation could be an interesting further study. We have added the study you mentioned as reference 28 and modified our discussion as per your comment. 

Original sentence 
First, we assumed that the variations in intrathoracic pressure during respiration did not significantly affect the carotid FTc. However, low VT mechanical ventilation is especially essential for patients with poor lung conditions, such as ARDS. Poor lung compliance and consequent large fluctuation of intrathoracic pressure may result in different results.

Revised sentence (Discussion section, Line 332-337)
First, we assumed that the variations in intrathoracic pressure during respiration did not significantly affect the carotid FTc. In addition, there were little differences between FTc6 and FTc10. However, a recent study showed that the respiratory cycle could induce a variation in FTc [28]. This could affect the results of our study. A study for the evaluation of the FTc variation during the respiratory cycle is needed. Further studies in patients with poor lung compliance or large intrathoracic pressure change may result in different results.

Point 3: Line 323 should reflect a large change in trans-pulmonary* pressure in severe pulmonary ARDS, rather than intrathoracic* pressure. Intrathoracic pressure (i.e., pleural pressure) would change more with extrapulmonary ARDS and/or obesity

Response 3
We appreciate your suggestion. We have modified the sentences in conjunction with point 2 as you suggested.

Original sentence 
However, low VT mechanical ventilation is especially essential for patients with poor lung conditions, such as ARDS. Poor lung compliance and consequent large fluctuation of intrathoracic pressure may result in different results.
Revised sentence (Discussion section, Line 333-337)
However, a recent study showed that the respiratory cycle could induce a variation in FTc [28]. This could affect the results of our study. A study for the evaluation of the FTc variation during the respiratory cycle is needed. Further studies in patients with poor lung compliance or large intrathoracic pressure change may result in different results.

Point 4: the format of table 3 makes it difficult to follow. I suggest making it more pleasing by spacing it better, this table contains very important information that should be easy for the reader to extract.

Response 4
We appreciate your suggestion. We have modified Table 3 accordingly.

Point 5: can the authors comment on limitations between Bazett and Wodey’s equations, for example how each of them handles extremes of heart rates? Is this a factor for the heart rates reported in the author’s work?

Response 5
It has been reported that the use of Bazett’s equation for calculating FTc could be affected by heart rates. Wodey’s equation can decrease the effect of heart rates on the calculation [7]. In our study, mean value of FTcW (329.9±26.3) measured at baseline was lower than that of FTcB (348.3±26.3) (P-value<0.001, paired t –test). One reason for this result could be the presence (5 of 36 patients) of the patients with tachycardia over 100bpm. Our study was performed in the patients clinically diagnosed with hypovolemic status. Therefore, we anticipated that the range of the heart rate could be wide so that the use of Wodey’s formula would yield more accurate results. However, the numbers of the patients in the grey zone of FTc6B and FTc6W were comparable (13 [36%] and 12 [33%], respectively). The calculation of FTc measured in the large size of patient group with wider ranges of heart rate could show significant differences between the accuracy of two equations.

Point 6: can the authors comment on the heart rate-to-respiratory rate ratio and how this affects pulse pressure variation as a measure of fluid responsiveness? Was this a factor here?

Response 6
Per your comments, pulse pressure variation could be affected by the ratio between heart rate and respiratory ratio [8]. We analyzed our patients’ data and found that it is hard to discriminate the effects of this factor to PPV because the heart rate-to-respiratory rate ratio was >3.6 at any point in our study.

Point 7: lines 241-243 and 346-350 appear to be vestiges from the journal’s manuscript template; please cut.

Response 7
We apologize for this confusion. It has been deleted per your comment.

References
1.     Kenny, J.-É.S.; Barjaktarevic, I.; Mackenzie, D.C.; Eibl, A.M.; Parrotta, M.; Long, B.F.; Eibl, J.K. Diagnostic characteristics of 11 formulae for calculating corrected flow time as measured by a wearable doppler patch. Intensive Care Medicine Experimental 2020, 8, 54.
2.    Jalil, B.; Thompson, P.; Cavallazzi, R.; Marik, P.; Mann, J.; El-Kersh, K.; Guardiola, J.; Saad, M. Comparing changes in carotid flow time and stroke volume induced by passive leg raising. The American journal of the medical sciences 2018, 355, 168-173.
3.      Doctor, M.; Siadecki, S.D.; Cooper, D., Jr.; Rose, G.; Drake, A.B.; Ku, M.; Suprun, M.; Saul, T. Reliability, laterality and the effect of respiration on the measured corrected flow time of the carotid arteries. The Journal of emergency medicine 2017, 53, 91-97.
4.    Kim, D.H.; Shin, S.; Kim, N.; Choi, T.; Choi, S.H.; Choi, Y.S. Carotid ultrasound measurements for assessing fluid responsiveness in spontaneously breathing patients: Corrected flow time and respirophasic variation in blood flow peak velocity. British journal of anaesthesia 2018, 121, 541-549.
5.    Michard, F. Changes in arterial pressure during mechanical ventilation. Anesthesiology 2005, 103, 419-428.
6.    Muller, L.; Louart, G.; Bousquet, P.J.; Candela, D.; Zoric, L.; de La Coussaye, J.E.; Jaber, S.; Lefrant, J.Y. The influence of the airway driving pressure on pulsed pressure variation as a predictor of fluid responsiveness. Intensive care medicine 2010, 36, 496-503.
7.    Mohammadinejad, P.; Hossein-Nejad, H. Calculation of corrected flow time: Wodey's formula vs. Bazett's formula. Journal of critical care 2018, 44, 154-155.
8.    De Backer, D.; Taccone, F.S.; Holsten, R.; Ibrahimi, F.; Vincent, J.L. Influence of respiratory rate on stroke volume variation in mechanically ventilated patients. Anesthesiology 2009, 110, 1092-1097.

Sincerely,

Reviewer 2 Report

I had the pleasure of reviewing a manuscript titled "Ability of carotid corrected flow time to predict fluid responsiveness in patients mechanically ventilated using low tidal volume after surgery" by Seungho Jung et al. 

The authors assess the operating characteristics of FTc in patients with low tidal volume breathing.  PPV is well validated, but is not accurate in patients with reduced tidal volume respiration.  The gold standard for fluid responsiveness was the response to 8 mL/kg of crystalloid as determined by FloTrac/Vigileo (surrogate for stroke volume).  The authors found FTc to be a better predictor of volume responsiveness than FFT, which makes physiologic sense. 
General Comments:
- Very well written paper
- Logical study design
- Authors excluded patients on vasopressors, which limits generalizability of the study. This is addressed in limitations.
- I suggest that authors describe how the FloTrac sensor functions, since many readers may not be familiar with it.  They should also cite a validation study that confirms it is accurate in predicting stroke volume. 

Author Response

Response to Reviewer 2 Comments

I had the pleasure of reviewing a manuscript titled "Ability of carotid corrected flow time to predict fluid responsiveness in patients mechanically ventilated using low tidal volume after surgery" by Seungho Jung et al.

The authors assess the operating characteristics of FTc in patients with low tidal volume breathing.  PPV is well validated, but is not accurate in patients with reduced tidal volume respiration.  The gold standard for fluid responsiveness was the response to 8 mL/kg of crystalloid as determined by FloTrac/Vigileo (surrogate for stroke volume).  The authors found FTc to be a better predictor of volume responsiveness than FFT, which makes physiologic sense.

General Comments:

- Very well written paper

- Logical study design

- Authors excluded patients on vasopressors, which limits generalizability of the study. This is addressed in limitations.

We would like to thank you for your time and effort in reviewing our manuscript and providing valuable comments, which have greatly helped us in revising the manuscript. We have carefully read your comments and have revised the manuscript. Our detailed responses to each comment are written below.

Point 1: I suggest that authors describe how the FloTrac sensor functions, since many readers may not be familiar with it. They should also cite a validation study that confirms it is accurate in predicting stroke volume.

Response 1

We sincerely appreciate your advice. We added a description of the Flotrac sensor in the Discussion section as well as a validation study as a reference. The changes are shown below.

Original sentences : The Flotrac/VigileoTM system is an uncalibrated device and showed inaccuracies in patients with severe arrhythmia, severe aortic valve regurgitation, and in those using high dose vasopressor [28].

Revised sentences (Discussion section, Lines 341-348): The Flotrac/VigileoTM system is a minimally invasive cardiac output monitoring system, which has a blood flow sensor connected to the patient’s arterial line. It calculates cardiac output by using individual demographic data and arterial pressure waveform analysis. Cardiac output measured by Flotrac showed acceptable agreement with thermodilution technique [29]. It is an uncalibrated device and showed inaccuracies in patients with severe arrhythmia, severe aortic valve regurgitation, and in those using high dose vasopressor [30].

References

Original reference

  1. Sangkum, L.; Liu, G.L.; Yu, L.; Yan, H.; Kaye, A.D.; Liu, H. Minimally invasive or noninvasive cardiac output measurement: An update. Journal of anesthesia 2016, 30, 461-480.

Revised references (References section, Lines 472-476)

  1. Mayer, J.; Boldt, J.; Poland, R.; Peterson, A.; Manecke, G.R., Jr. Continuous arterial pressure waveform-based cardiac output using the Flotrac/Vigileo: A review and meta-analysis. Journal of cardiothoracic and vascular anesthesia 2009, 23, 401-406.
  2. Sangkum, L.; Liu, G.L.; Yu, L.; Yan, H.; Kaye, A.D.; Liu, H. Minimally invasive or noninvasive cardiac output measurement: An update. Journal of anesthesia 2016, 30, 461-480.

Sincerely,

Reviewer 3 Report

Authors reported that  carotid FTc can predict fluid responsiveness better than PPV during low VT ventilation. 

1) This paper was performed in the very small number of patients. Therefore, larger number of patient is required to analyse.

2) Authors should describe the exclusion criteria of patients in more detail. 

Author Response

Response to Reviewer 3 Comments

Authors reported that carotid FTc can predict fluid responsiveness better than PPV during low VT ventilation.

We would like to thank you for your time and effort in reviewing our manuscript and providing valuable comments, which have greatly helped us in revising the manuscript. We have carefully read your comments and have revised the manuscript. Our detailed responses to each comment are written below.

Point 1: This paper was performed in the very small number of patients. Therefore, larger number of patient is required to analyse.

Response 1: As you correctly pointed out, we planned a small sample size for this study calculated by the power analysis. Our study was performed by a single ultrasonography examiner during his own duty in real clinical condition. If we enrolled a larger number of patients, the results of our study could be more confirmative and further subgroup analysis could also have been possible. This is the limitation of our study. We strongly agree with your comment that a larger clinical trial is required. We additionally described this limitation in the Discussion as per your comment.

Original sentences

Therefore, it has been used to monitor the cardiac output in several studies [27,30,31]. Finally, the measurement of FTc in our study was performed manually by two pre-trained experts using captured images.

Revised sentences (Discussion section, line number 352-356)

Therefore, it has been used to monitor the cardiac output in several studies [27,32,33]. Third, this study was conducted in a small population. Our study was performed by a single ultrasonography examiner in a surgical ICU. This limited the enrollment of a large number of patients. Our findings need to be reproduced in a study with a larger number of subjects. Finally, the measurement of FTc in our study was performed manually by two pre-trained experts using captured images.

Point 2: Authors should describe the exclusion criteria of patients in more detail.

Response 2: We have added more details of the exclusion criteria as per your comment.

Original sentences

The exclusion criteria were: body mass index of >40 or <15 kg/m2, common carotid artery stenosis of >50% which was previously diagnosed or newly detected during the study period, cardiac rhythm other than sinus rhythm, valvular heart disease, left ventricular ejection fraction of <50%, right ventricular failure, moderate to severe chronic obstructive pulmonary disease, pulmonary hypertension, increased intracranial pressure, and patients who needed vasopressor infusion to maintain normal blood pressure before being transferred to the ICU.

Revised sentences (2.1. Study population section, line number 82-97)

The exclusion criteria were as follows: body mass index of >40 or <15 kg/m2, common carotid artery stenosis of >50% which was previously diagnosed (by conventional angiography, computed tomographic angiography, magnetic resonance angiography or duplex ultrasonography) or newly detected during the study period, cardiac rhythm other than sinus rhythm observed during the study period, moderate to severe valvular heart disease detected by preoperative echocardiography, left ventricular ejection fraction of <50%, right ventricular failure, moderate to severe chronic obstructive pulmonary disease, pul-monary mean arterial pressure of >25mmHg, suspected or diagnosed increased intracranial pressure, pregnancy, patients who were unable to read the consent form (e.g. illiterate, foreigner, etc.), need for vasopressor infusion to maintain normal blood pressure before being transferred to the ICU. Furthermore, we excluded patients whom the ultrasound probe could not reach the carotid artery because of an overlapping position of the surgical wound and exam site.

Sincerely,

Round 2

Reviewer 1 Report

The authors have addressed the comments/criticisms well; their work is important and interesting. The authors may find this recent publication interesting.  

https://bjanaesthesia.org/article/S0007-0912(21)00298-1/fulltext

Reviewer 3 Report

The revised paper has been improved.